

# Curcumin improves atrial fibrillation susceptibility by regulating tsRNA expression in aging mouse atrium

Xueshan Luo[1,2,3,4,*], Panyue Liu[1,2,4,*], Xingdong Ye[1,2], Jintao He[1,2,4], Yingyu Lai[1,2], Yidong Lv[1,2], Xiongbin Wu[5], Yang Liu[1,2], Qianhuan Zhang[1,2], Hui Yang[1,2], Wei Wei[1,2], Chunyu Deng[1,2,4], Sujuan Kuang[1,2], Shulin Wu[1,2], Yumei Xue[1,2,4] and Fang Rao[1,2,4]

[1] Guangdong Provincial Key Laboratory of Clinical Pharmacology, Guangdong Provincial People's Hospital, Guangdong Academy of Medical Sciences, Southern Medical University, Guangzhou, Guangdong, China
[2] Guangdong Cardiovascular Institute, Guangzhou, Guangdong Provincial People's Hospital, Guangdong Academy of Medical Sciences, Guangdong, China
[3] Zhujiang Hospital, Southern Medical University, Guangzhou, Guangdong, China
[4] South China University of Technology, Guangzhou, Guangdong, China
[5] Jiangmen Central Hospital, Jiangmen, Guangdong, China
* These authors contributed equally to this work.

Corresponding authors
Yumei Xue, xueyumei@gdph.org.cn
Fang Rao, raofang@gdph.org.cn

## ABSTRACT

Age is an independent risk factor for atrial fibrillation (AF), and curcumin can delay aging related disease through reducing oxidative stress and inflammation. However, its target in aging-related AF remains unclear. Transfer RNA-derived small RNA (tsRNA) is a novel short non-coding RNA (sncRNA), and exerts a potential regulatory function in aging. This study was to explore the therapeutic targets of curcumin in atrium of aged mice by PANDORA-seq. Aged mice (18 month) were treated with curcumin (100 mg/kg). Rapid transjugular atrial pacing was performed to observe AF inducibility. SA-β-gal staining, reactive oxygen species (ROS) detection and qRT-PCR were used to assess the degree of aging and oxidative stress/inflammation levels. PANDORA-seq was performed to reveal the differentially expressed sncRNAs in the atrium of mice. The results showed that curcumin reduced the susceptibility AF of aged mice by improving aging-related atrial fibrosis. Compared to young mice (5 month) group, aged mice yielded 473 significantly altered tsRNA sequences, while 947 tsRNA sequences were significantly altered after treated with curcumin. Enrichment analysis revealed that the target genes were mainly related to DNA damage and protein modification. Compared with the 5 month group, the expression levels of mature-mt_tRNA-Val-TAC_CCA_end, mature-mt_tRNA-Glu-TTC_CCA_end, and mature-tRNA-Asp-GTC_CCA_end were up-regulated in the 18 month group, while the expression of mature-mt_tRNA-Thr-TGT_5_end was down-regulated. This trend was reversed in the 18 month + curcumin group. Increased cellular ROS levels, inflammation expression and senescence in aged mice atrium were improved by the down-regulation of mature-mt_tRNA-Val-TAC_CCA_end. In conclusion, our findings identified mature-mt_tRNA-Val-TAC_CCA_end participated in the mechanism of aging-related atrial fibrosis, providing new intervention target of aging-related AF.

# INTRODUCTION

Atrial fibrillation (AF) is one of the most common clinical arrhythmias, and it can lead to stroke, heart failure, and all-cause death (*Zhang et al., 2018*). Age is an important risk factor for AF. Epidemiological studies have shown that the prevalence of AF in the general population is about 1–2%, but it is higher than 15% in people over 80 years old (*Seccia et al., 2019*). Due to the aging population, the burden of AF is rapidly increasing worldwide and is expected to at least triple over the next 50 years (*Go et al., 2001*). The management of AF, while improved with pharmacological and invasive strategies, faces notable shortcomings. These include variable drug responses (*Van Gelder et al., 2002*), bleeding risks with anticoagulants (*Steffel et al., 2018*), procedural complications (*Parameswaran, Al-Kaisey & Kalman, 2021*), and diagnostic challenges, particularly in detecting asymptomatic cases (*Rosero et al., 2013*). Current risk stratification models incorporate factors such as age, gender, hypertension, and heart failure (*Shahid & Lip, 2017*). However, these models lack the precision required for individualized predictions. To address these limitations and enhance AF care, future research should explore innovative approaches, such as the identification of novel molecules like transfer RNA-derived small RNAs (tsRNAs), as discussed in this study. Personalized medicine and precision approaches may hold the key to more effective AF management in the coming years. Therefore, it is imperative to investigate the molecular mechanisms of aging-related AF.

Aging plays an important role in atrial electrical and structural remodeling, leading to increased susceptibility to AF. Triggers and mediators of aging, including DNA damage, oxidative stress, inflammation and mitochondrial damage, have been shown to contribute to AF progression (*Arreal et al., 2020*). Curcumin is a natural polyphenolic compound extracted from turmeric. Due to its hydrophobicity, it can diffuse through cell membranes and reach the endoplasmic reticulum, mitochondria, and nucleus, acting as a natural free radical scavenger (*Kim & Clifton, 2018*). A large amount of evidence have shown that the effects of curcumin on cardiovascular diseases are mediated through multiple mechanisms such as oxidative stress, inflammation, and cell death (*He et al., 2015*). In addition, curcumin can affect the key proteins of organism longevity through epigenetic methods, such as inhibiting protein histonedeacetylases (HDAC) activity, inducing the reversal of DNA methylation, and upregulating multiple miRNAs (*Hassan et al., 2016*). Therefore, curcumin has shown promising potential in the treatment of aging-related AF, but its drug target remains unclear.

Epigenetic alteration is considered as a hallmark of aging (*Li et al., 2020b*; *Lopez-Otin et al., 2013*), including mechanisms mediated by DNA methylation, histone modifications, chromatin remodeling, and noncoding RNAs (*Ling & Ronn, 2019*; *Yu et al., 2020*). TsRNA is a new class of sncRNA, which can be roughly divided into two categories: tiRNAs (or tRNA halves) and tRFs (tRNA-derived fragments). Recent new evidence from mammalian species highlighted that multiple biological functions were mediated by tsRNAs, including regulation of ribosomal biogenesis, translation initiation, tight control of retrotransposon

activity and epigenetics (*Shi et al., 2019*), which is associated with the key cellular processes such as self-renewal, differentiation and proliferation (*Cao, Cowan & Wang, 2020*). TsRNAs in the peripheral circulation are sensitive to acute inflammation (*Zhong et al., 2019*), aging (*Kim & Lee, 2019*) and tissue damage (*Xie et al., 2020*), and may potentially be used as non-invasive biomarkers. High-throughput sequencing technologies have been applied in degenerative disease research, which promoted the discovery of many dysregulated tsRNAs and hold promise for the development of future therapies. However, preliminary studies have not highlighted the function of tsRNAs as regulators of the aging process.

In this study, we first determined that curcumin reduced susceptibility to AF in aged C57BL/6 mice. Subsequently, PANDORA-seq was used to explore the differentially expressed sncRNAs in the atrium of young and aged mice treated with or without curcumin. The differential tsRNAs were then selected for bioinformatics analysis to explore their potential roles in this process. In particular, the novel mature-mt_tRNA-Val-TAC_CCA_end was identified as a target tsRNA, which was regulated by curcumin in the anti-aging process of the atrium in aged mice. In summary, the results provide insights into the molecular mechanisms of tsRNAs in aging atrium and reveal potential drug targets for curcumin in the treatment of aging-related AF.

# METHODS AND MATERIALS

## Animals

All procedures and protocols in this study have been approved by the animal experiment ethics committee of Guangdong Provincial People's Hospital, Guangdong Academy of Medical Sciences (No. GDREC2016128A). All animals received care in compliance with the Guide for the Care and Use of Laboratory Animals published by the US National Institutes of Health (NIH publication no. 85-23, revised 1996). Male C57BL/6 mice were purchased from Guangzhou University of Traditional Chinese Medicine Laboratory Animal Center under the license number SCXK2018-0034. The mice were housed at $22 \pm 2$ °C with a humidity of $55 \pm 10\%$ and a 12 h light/dark cycle with adibitum access to food and water. They were randomly divided into three groups, namely the 5 month group, the 18 month group, and the 18 month + curcumin group ($n = 6$). Curcumin (100 mg/kg/d) was administered once daily from the age of 12 to 18 months *via* food feeding. To reduce the pain during the experiment, the mice were sacrificed by cervical dislocation after anesthesia by intraperitoneal injection of pentobarbital sodium, and the hearts were dissected to collect the atrial appendages, which were immediately placed into a liquid nitrogen tank, and then transferred to a freezer at −80 °C for later inspection.

## Rapid transjugular atrial pacing

The mice were weighed, anesthetized by intraperitoneal injection of 1% pentobarbital sodium (60 mg/kg), and fixed on a heating pad (RWD Life Science Company, Shenzhen, China) at a constant temperature ($37 \pm 1$) °C. At least six mice from each group were tested. Surface ECG and intracavitary ECG were recorded using the iWorx system

(Dover, NH, USA). A sheath was placed in the right jugular vein through a small incision, and the electrode catheter was inserted along the sheath to the right atrium. After recording the intracavity electrogram, rapid pacing of the endocardium was performed. The pacing threshold was measured and the atrial burst stimulation (short burst electrical stimulation) mode was used to induce AF and record its duration, sinus node recovery time (SNRT), and other indicators. On the ECG, AF is characterized by the disappearance of P waves and the appearance of f waves; the restoration of sinus rhythm is denoted by the recurrence of P waves and the disappearance of f waves. The duration of AF was defined as the time from AF occurrence to AF termination.

After completing the electrophysiological experiment, the mice were sacrificed by cervical dislocation after deep anesthesia. The hearts were dissected, and the atrial appendages were collected for histological examination and sequencing.

## PANDORA-seq

### Small RNA library construction and sequencing

Small RNA library construction and sequencing were performed by Epibiology Ltd. Three biological replicates were set for each group. Total RNA was extracted from the atrium using TRIZOL reagent (1 ml; Accurate Ciotechnology, Hunan, China). Total RNA (200 ng) was added to 50 µl reaction mixture containing 4 µg/mL AlkB (Epibiotek, Incheon, Republic of Korea) at 37 °C for 30 min. The mixture was then transferred to another tube of reaction mixture, which containing 10U T4PNK (New England Biolabs), at 37 °C for 20 min. These two steps were necessary for dephosphorylation and demethylation of RNA fragments in the 15–50 nt region, while converting 3′ phosphate or 2′, 3′ ring phosphate into 3′-OH. Converting 5′-OH to 5′ phosphate resolves adapter ligation issues due to end modifications. RNA fragments were separated by PAGE, and 15-45 nucleotide bands were selected for recovery. These adapters were obtained from the QIAseq® miRNA Library Kit (331505; Qiagen, Hilden, Germany) and ligated sequentially. Expanded flow cells were sequenced by Epibiotech (Guangzhou Wai Biotech, Guangzhou, China) using the SE75 system. The statistical power of this experimental design, calculated in RNASeqPower is 0.80 (https://rodrigo-arcoverde.shinyapps.io/rnaseq_power_calc/).

## Small RNA annotation

Small RNA sequences were annotated using the pipeline SPORTS (a small non-coding RNA annotation pipeline optimized for rRNA and tRNA-derived small RNAs, https://github.com/junchaoshi/sports1.1). Small RNA annotation was performed for annotating miRNAs, tRNAs, rRNAs, and other small non-coding RNAs from TIGR, miRBase, tRFdb, MINTbase, and piRBase. Subsequently, the expression difference of sncRNAs between the 5, 18 and 18 month+curcumin group was analyzed. TsRNA reads were normalized to RPM (reads per million). Differentially expressed tsRNAs were detected by using the R package edgeR. TsRNAs were considered significantly differentially expressed when the $P$-value was <0.05 and $|\log_2$-fold change$| \geq 1$.

## Target mRNA prediction and correlation analysis of differential tsRNA

To verify the reliability of the PANDORA-seq, we numbered tsRNAs containing the ends in turn and randomly selected four numbers. Results extracted four tsRNA containing ends as mature-mt_tRNA-Val-TAC_CCA_end, mature-mt_tRNA-Glu-TTC_CCA_end, and mature-tRNA-Asp-GTC_CCA_end. To further reveal the functions of differential tsRNAs, we performed a bioinformatics analysis of these selected tsRNA. miRanda (http://www.microrna.org) and RNAhybrid (https://bibiserv.cebitec.uni-bielefeld.de/) were used to predict differentially expressed target mRNAs (screening criteria: miRanda, score ≥150, energy <−20). The miRanda score is the combined score of tsRNA and its target gene; a higher score indicates more accurate target gene prediction. Based on the regulatory relationship between tsRNA and predicted target genes, the tsRNA and target gene network were constructed using Cytoscape software (version 3.7.2, https://cytoscape.org/). Furthermore, enrichment analysis was performed on the target genes of tsRNAs on metascape (http://metascape.org/). Smaller $P$ values indicated more significant effects, and terms with $P < 0.05$ were considered statistically significant.

## Cell culture and mock transfection

Mouse atrial fibroblasts (MAF) were used in the study. The cells were obtained from the atrial tissues of 12 mice and were isolated by 50 mL 0.25% trypsin-EDTA (GIBCO, Grand Island, NY, USA) digestion. The cells were then cultured at 37 °C in DMEM F12 medium (GIBCO, Grand Island, NY, USA) containing 10% FBS, 100 U/ml penicillin, and 100 U/ml streptomycin. In order to induce senescence, passage 4 (P4) MAF was treated with 100 µM Tert-Butyl hydroperoxide (TBHP) for 2 h, then replaced with normal medium, and cultured for 2 days, while the control group was treated with PBS. To knock down and overexpress mature-mt_tRNA-Val-TAC_CCA_end, cells were seeded into 6-well plates 24 h before transfection with Lipofectamine 3000 (Invitrogen, Waltham, MA, USA), which was performed according to the manufacturer's instructions. A negative control (GenePharma, Shanghai, China) at a concentration of 1 nM and tsRNA mimics or inhibitors (GenePharma, Shanghai, China) at a concentration of 1 nM were used in transfection. Eight hours after transfection, the cells were used for further experiments.

## Tissue and cell senescence β-gal staining

The kit (#9860) from Cell Signaling Company was used for SA-β-gal staining according to the manufacturer's instructions. The activity of β-galactosidase, a marker of senescent cells, was detected in a solution with a pH value of 6. Senescent tissues/cells show a green stain under acidic conditions.

## ROS detection

After transfection or treated with TBHP, MAFs were incubated with 10 µM DCFH-DA (Beyotime, Shanghai, China) at 37 °C for 20 min, then washed three times with a serum-free cell culture medium. The intracellular ROS content, the green fluorescence intensity in the cells was detected using a fluorescent confocal microscope under 488 nm

excitation wavelength and 525 nm emission wavelength. The stronger the fluorescence, the higher the ROS content.

## Validation by qRT-PCR

qRT-PCR was performed on mouse atrial tissue ($n$ = 6) and atrial fibroblasts ($n$ = 3) to validate tsRNA. All experiments were performed in biological replicates, each in technical triplicates for the groups. AG RNAex Pro RNA Reagent (Accurate Ciotechnology, Hunan, China) was used to extract the total RNA of the atrium. Total RNA was reverse transcribed using the miRNA 1st Strand cDNA synthesis kit (Accurate Ciotechnology, Hunan, China). qRT-PCR was performed using 10 μl 2×SYBR Green Pro TaqHS Premix II (Accurate Ciotechnology, Hunan, China), cDNA (40 ng), 0.8 μl miRNA-specific (10 μM), 0.8 μl miRNA qRT-PCR 3′ primer (10 μM) and RNase free water, with a final volume of 20 μl. The reaction mixture of the sample was incubated in the StepOne real-time PCR system (Applied Biosystems, Foster City, CA, USA) at 95 °C for 10 min, 95 °C, 15 s, 60 °C, 1 s, and finally at 72 °C for 30 s, for 35 PCR cycles. Normalization was performed using U6 small nuclear RNA. All analyses were performed with the $2^{-\Delta\Delta Cq}$ method. The primer sequences used for qRT-PCR are listed in Table 1.

## Statistical analysis

Results are expressed as mean ± standard error of the mean (SEM). Statistical analysis was performed by SPSS 23.0 software. Differences between the two groups were determined using Student's t-test. One-way analysis of variance (ANOVA) was used for comparison among the three groups. $P < 0.05$ was defined as statistically significant.

# RESULTS

## Curcumin reduced susceptibility to AF by attenuating atrial aging and inflammation in aged mice

Rapid atrial pacing was performed through the jugular vein to observe the inducible rate of AF (Fig. 1A) and the electrophysiological characteristics of mice (Table 2). Compared with the 5 month mice, a significantly prolonged PR interval was observed in 18 month mice ($P < 0.05$), and the sinus node recovery time (SNRT) and rate-corrected sinus node recovery time (cSNRT) were also significantly increased ($P < 0.05$). Curcumin treatment significantly improved the above indicators. Furthermore, the high induction rate of AF in mice in the 18 month group was significantly improved by curcumin (100 mg/kg/d) treatment (Fig. 1B). SA-β-gal staining showed greater green staining in the atrium of 18 month mice than in the 5 month group, suggesting a significantly higher degree of aging in aged mice, which could be improved by curcumin treatment (Fig. 1C). In addition, qRT-PCR showed that curcumin also reduced the inflammatory index IL-6/TNFα in the atrial tissue of 18 month mice (Fig. 1D). Therefore, these results suggest that curcumin might ameliorate the atrial tissue aging through decreasing the inflammatory response, thereby reducing the susceptibility of AF in aged mice.

**Table 1 The primers for RT-qPCR used in present study.**

| Gene | Primer sequence (5′–3′) |
|---|---|
| Mature-mt_tRNA-Val-TAC_CCA_end | Forward:ACACCCAGAAGATTTCATGACCAATGAACACTCTGACCA |
| | Reverse:Provided by Accurate CiotechnologyCo.,Ltd. |
| Mature-mt_tRNA-Glu-TTC_CCA_end | Forward:CATGTCATTGGTCGCAGTTGAATGCTGTGTAGAAATACCA |
| | Reverse:Provided by Accurate CiotechnologyCo.,Ltd. |
| Mature-tRNA-Asp-GTC_CCA_end | Forward:ACGCGGGAGACCGGGGTTCGATTCCCCGACGGGGAGCCA |
| | Reverse:Provided by Accurate CiotechnologyCo.,Ltd. |
| Mature-mt_tRNA-Thr-TGT_5_end | Forward:GTCTTGATAGTATAAAC |
| | Reverse:Provided by Accurate CiotechnologyCo.,Ltd. |
| U6 | Forward:GGAACGATACAGAGAAGATTAGC |
| | Reverse:TGGAACGCTTCACGAATTTGC |
| Smad3 | Forward:CCAGATGAACCACAGCATGGA |
| | Reverse:CTACTGTCATGGACGGCTGTGAA |
| Mlk1 | Forward:GAGCCAGTTCAATTCGACCTCC |
| | Reverse: GGCTAGTTTCTTCCTCGCAGT |
| Ndufs6 | Forward:AGATCACGCATACCGGCCA |
| | Reverse:CAATGGCAAAGTTCTCATTCACCT |
| Mrpl27 | Forward:GAAGTCTACGTGCCCAATCCC |
| | Reverse:TTTGAAGGTTCCCTCCGGTT |
| IL6 | Forward:CCACTTCACAAGTCGGAGGCTTA |
| | Reverse:CCAGTTTGGTAGCATCCATCATTTC |
| TNF-α | Forward:ACTCCAGGCGGTGCCTATGT |
| | Reverse:GTGAGGGTCTGGGCCATAGAA |
| GAPDH | Forward:GGCACAGTCAAGGCTGAGAATG |
| | Reverse:ATGGTGGTGAAGACGCCAGTA |

## Exemplary annotation and profiling of PANDORA-seq datasets generated by SPORTS1.1

SPORTS1.1 was used to annotate and analyze the PANDORA-seq data sets, and the sncRNA expression profiles of the atrium tissue in the 5, 18, or 18 month mice treated with 100 mg/kg/d curcumin were shown in Fig. 2. It displayed the length distribution for sncRNAs. A large number of modified sncRNAs were identified, including miRNAs,

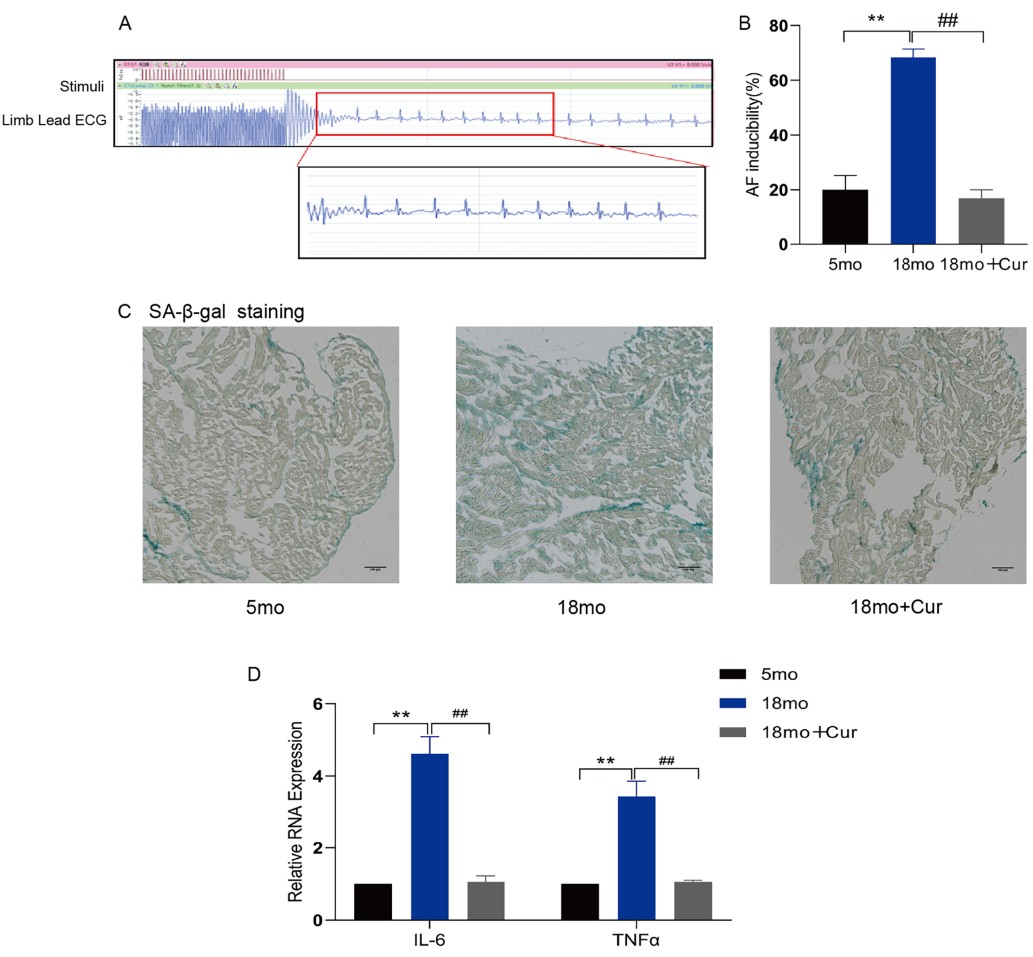

**Figure 1 Curcumin reduces the susceptibility to AF by attenuating aging and inflammation in aged atrium.** (A) AF spontaneously reverts to a typical surface ECG recording with Sinus rhythm (SR) and a typical atrial wave (f wave). (B) Statistical histogram of AF induction rate. (C) The representative images of SA-β-gal staining of atrial tissue in the 5 month group, the 18 month group and the 18 month + curcumin group. The green staining of the tissue indicates senescence. Scale bar, 100 μm. (D) The expression of inflammation indicators of the three groups. $n = 6$, **$P < 0.01$ vs. 5 month group, ##$P < 0.01$ vs. 18 month group.

piRNAs, tsRNAs, rsRNAs, and some unannotated sncRNAs, whose lengths were concentrated in the range of 15–45 nt. However, tsRNA and rsRNA accounted for the largest proportions of these sncRNAs. The role of sncRNA in the aging process was speculated to be mainly mediated by tsRNA and rsRNA, so further experiments were performed on tsRNA.

## General characteristics of tsRNAs expression

The tsRNA expression profiles were annotated and analyzed. According to the source of tRNA region, tsRNA can be divided into four groups: 3′ end, 5′ end, CCA end, and other tsRNAs (Figs. 3A–3C). In addition, tsRNAs are mainly derived from tRNAs of glycine (Gly), serine (Ser), valine (Val), and glutamine (Glu) (Figs. 3D–3F). Compared with the 5

**Table 2 Electrophysiology data.**

|  | 5 month | 18 month | 18 month + Cur |
|---|---|---|---|
| N | 6 | 6 | 6 |
| BW (g) | 26.33 ± 0.76 | 33.12 ± 1.41** | 33.97 ± 0.84 |
| HR (BPM) | 439.91 ± 34.59 | 413.33 ± 39.84 | 395.74 ± 26.37 |
| PWD (ms) | 15.83 ± 0.75 | 20.17 ± 0.65** | 16.50 ± 0.428## |
| PR interval (ms) | 44.83 ± 0.70 | 53.50 ± 2.17* | 42.67 ± 0.84## |
| QRS duration (ms) | 14.00 ± 0.45 | 15.17 ± 1.45 | 13.83 ± 1.22 |
| QT inerval (ms) | 28.33 ± 1.12 | 30.33 ± 1.86 | 28.50 ± 1.31 |
| SNRT (ms) | 134.50 ± 4.11 | 231.17 ± 22.00* | 185.50 ± 8.48 |
| CSNRT (ms) | 25.82 ± 2.18 | 79.67 ± 11.82* | 33.01 ± 6.66# |

**Note:**
HR, heart rate; PWD, $p$ wave duration; SNRT, sinus node recovery time; CSNRT, corrected SNRT. Data are mean ± SEM; * < 0.05, ** < 0.01 *vs*. 5 m. # < 0.05, ## < 0.01 *vs*. 18 m.

month group, a higher total tsRNA expression was observed in the 18 month group, while a reversed trend was found in the 18 month + curcumin group with a different expression profile (Figs. 3A–3C). Moreover, significant differences in tsRNA origin sites were found among the three groups. Compared with the 5 month group, the total tsRNA expression in the 18 month group was up-regulated, while curcumin treatment reversed this trend and changed the expression profile (Figs. 3G–3I).

## TsRNA expression difference and target gene prediction

Hierarchical clustering heatmaps and volcano plots were used to visualize tsRNA divergence (Figs. 4A–4C). The 5 month group was compared to the 18 month group, yielding 473 significantly altered tsRNA sequences (212 down-regulated, 261 up-regulated), from 127 kinds of tsRNA species ($P < 0.05$, $|\log_2 FC| \geq 2$), while 947 tsRNA sequences were significantly altered between the 18 month group and the 18 month +curcumin group (647 tsRNAs down-regulated, 300 up-regulated), from 154 kinds of tsRNA species ($P < 0.05$, $|\log_2 FC| \geq 2$). Overall, 99 common differentially expressed tsRNA sequences (12 kinds of tsRNA species) between the pairwise comparisons in 5, 18 and 18 month + curcumin groups, of which 52 tsRNA sequences from mt-tRNAs and 47 tsRNA sequences from tRNAs (Fig. 4D, Table 3).

To reveal the function of tsRNAs, we predicted potential target mRNAs of tsRNAs (mature-mt_tRNA-Val-TAC_CCA_end, mature-mt_tRNA-Thr-TGT_5_end, mature-mt_tRNA-Glu-TTC_CCA_end, mature-tRNA-Asp-GTC_CCA_end) using miRanda and RNAbrid databases. A total of 102 mRNAs were associated with the 4 tsRNAs. The data was then acquired into Cytoscape to create a brand-new tsRNA-mRNA network to visualize and showed that one tsRNA might be associated with 0 or more mRNAs. Mature-mt_tRNA-Val-TAC_CCA_end was matched with 46 mRNAs, mature-mt_tRNA-Glu-TTC_CCA_end was matched with 17 mRNAs and mature-tRNA-Asp-GTC_CCA_end was matched with 39 mRNAs (Fig. 4E). It was noteworthy to mention that mature-mt_tRNA-Thr-TGT_5_end could not match with target mRNAs (Fig. 4E). Furthermore,
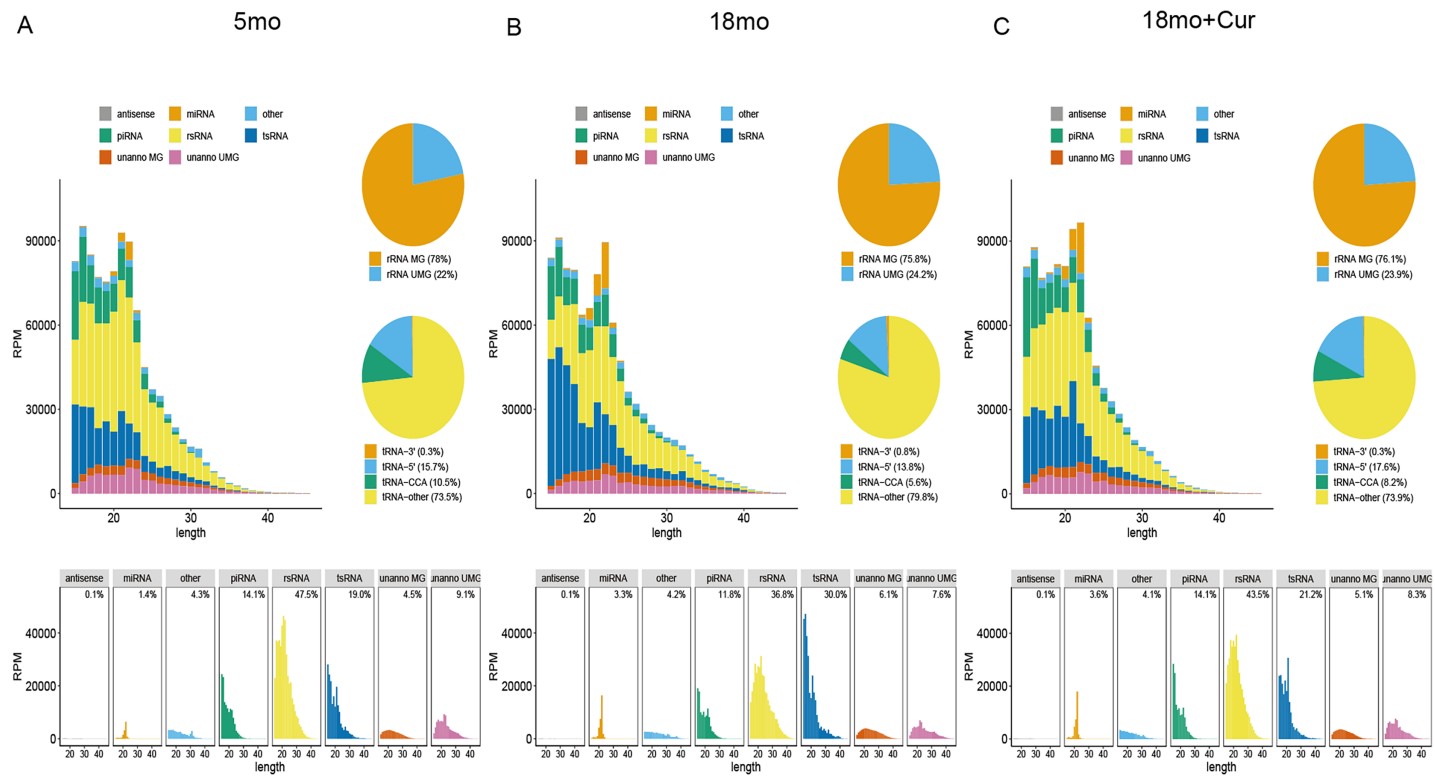

**Figure 2 Exemplary annotation and profiling of PANDORA-seq datasets generated by SPORTS1.1.** (A) Categorization and length distribution analysis of different sncRNA types in the 5 month group. (B) Categorization and length distribution analysis of different sncRNA types in the 18 month group. (C) Categorization and length distribution analysis of different sncRNA types in the 18 month+curcumin group. RPM, reads per million. Unanno, unannotated. MG, matched genomes. UMG, unmatched genomes.

enrichment analysis was performed on these tsRNA target genes on metascape (http://metascape.org/), revealing that the target genes of these 4 tsRNAs were mainly involved in DNA damage, protein modification, nutrient metabolism, and other processes, including DNA damage response (U-WP707), R-SUMOylation of transcription cofactors (R-HSA-3899300), and 2-Oxocarboxylic acid metabolism (hsa01210), *etc.*, (Fig. 4F).

## qRT-PCR verification of differential tsRNA and target genes

To verify the reliability of the PANDORA-seq, we confirmed the expression of tsRNAs and their target genes in these three groups using qRT-PCR. We found that, compared with the 5 month group, the expression of mature-mt_tRNA-Thr-TGT_5_end was down-regulated, and mature-mt_tRNA-Val-TAC_CCA_end, mature-mt_tRNA-Glu-TTC_CCA_end, and mature-tRNA-Asp-GTC_CCA_end were up-regulated in the 18 month group, which could be improved by the treatment of curcumin (Figs. 5A–5D). Their relative expression levels were the same as those obtained by random sequencing. Compared with the 5 month group, the expression of mature-mt_tRNA-Val-TAC_CCA_end core target genes (Smad3, Mkl1, Ndufs6, Mrpl27) was increased in the 18 month group, which was reversed by curcumin treatment excepting for Mrpl27 (Figs. 5E–5H).

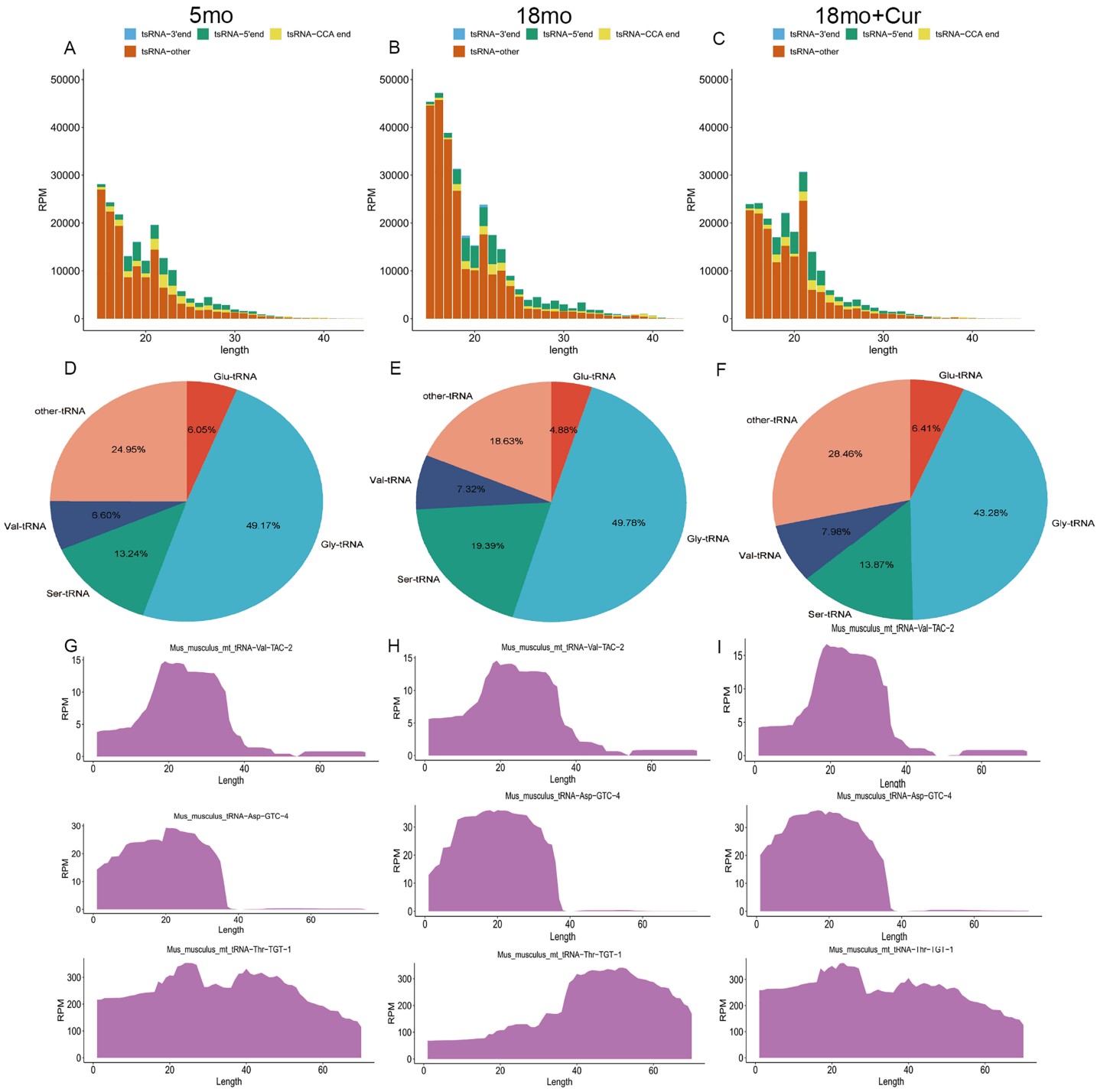

**Figure 3** **General characteristics of tsRNAs expression.** (A) TsRNA expression profiles in the 5 month group. (B) TsRNA expression profiles in the 18 month group; (C) TsRNA expression profiles in the 18 month + curcumin group. (D) TsRNAs derived from tRNAs containing different amino acids in the 5 month group. (E) TsRNAs derived from tRNAs containing different amino acids in the 18 month group. (F) TsRNAs derived from tRNAs containing different amino acids in the 18 month+curcumin group. (G) Sedimentation coefficient of tRNA at the tsRNA origin site in the 5 month group. (H) Sedimentation coefficient of tRNA at the tsRNA origin site in the 18 month group. (I) Sedimentation coefficient of tRNA at the tsRNA origin site in the 18 month+curcumin group.

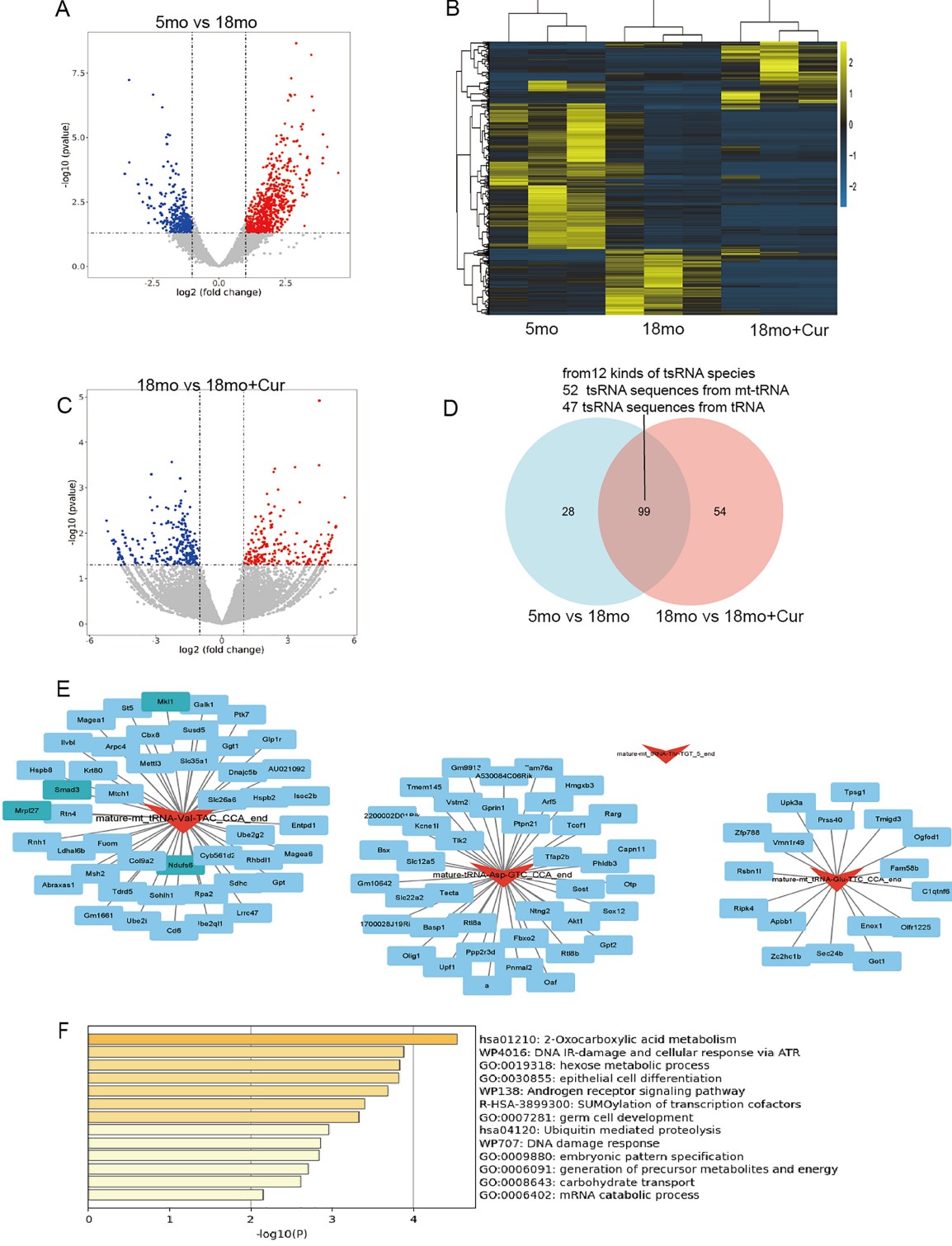

**Figure 4 Ts RNA expression difference and target gene prediction.** (A) Volcano plots of tsRNA sequence differences between 5 and 18 month group. (B) Hierarchical clustering heatmaps of tsRNA sequence differences among 5, 18 and 18 month + curcumin group. (C) Volcano plots of tsRNA sequence differences between 18 and 18 month + curcumin group. Blue indicates low expression levels and red indicates high expression levels. (D) The predicted target genes from the miRanda and RNAhybrid databases were intersected using a Venn diagram. (E) Co-analysis networks between tsRNAs and their target mRNAs. (F) Target gene enrichment analysis graph.

**Table 3 The common differentially expressed tsRNA species in 5, 18 and 18 month + curcumin groups.**

| tsRNA | Sequence | 5 month *vs.* 18 month | | 18 month *vs.* 18 month + Cur | |
|---|---|---|---|---|---|
| | | log2FC | Regulation | log$_2$FC | Regulation |
| mature-mt_tRNA-Glu-TTC_5_end | GTTTCTGTAGTTGAATTAC | 1.5258 | Down | −1.9815 | Up |
| mature-mt_tRNA-Ile-GAT | AGAGTAAATTATAGAGGTTC | 1.5339 | Down | −1.725 | Up |
| mature-mt_tRNA-Thr-TGT_5_end | GTCTTGATAGTATAAAC | 2.3454 | Down | −1.9529 | Up |
| mature-mt_tRNA-Val-TAC | ATTAAAGCATCTGGCCTAC | 2.4118 | Down | −3.368 | Up |
| mature-mt_tRNA-Arg-TCG | TTAAAAAAAATTAATGATTTCGACTCATTAGATTATG | −1.5782 | Up | 2.3751 | Down |
| mature-mt_tRNA-Asn-GTT | GTAATAGGGTATTTAGCTGTTAACTAAATTTTCGTAGGTT | −2.4361 | Up | 3.8851 | Down |
| mature-mt_tRNA-Glu-TTC_CCA_end | CATGTCATTGGTCGCAGTTGAATGCTGTGTAGAAATACCA | −1.5546 | Up | 2.7139 | Down |
| mature-mt_tRNA-His-GTG | TACAAAAAACATTAGACTGTGAATCTGACAACAGGAAATAA | −2.4193 | Up | 3.3582 | Down |
| mature-mt_tRNA-Met-CAT | CTAATTAAGCTATCGGGCCCATACCCCGAAAACGTTGGTT | −2.2347 | Up | 3.898 | Down |
| mature-mt_tRNA-Val-TAC_CCA_end | ACACCCAGAAGATTTCATGACCAATGAACACTCTGACCA | −1.4414 | Up | 2.4784 | Down |
| mature-tRNA-Asp-GTC_CCA_end | ACGCGGGAGACCGGGGTTCGATTCCCCGACGGGGAGCCA | −2.2692 | Up | 2.6139 | Down |
| mature-tRNA-Gly-GCC | TGGTTCAGTGGTAGAATTCTCGCCTGCCACGCGGGAGG | −2.5269 | Up | 3.9017 | Down |

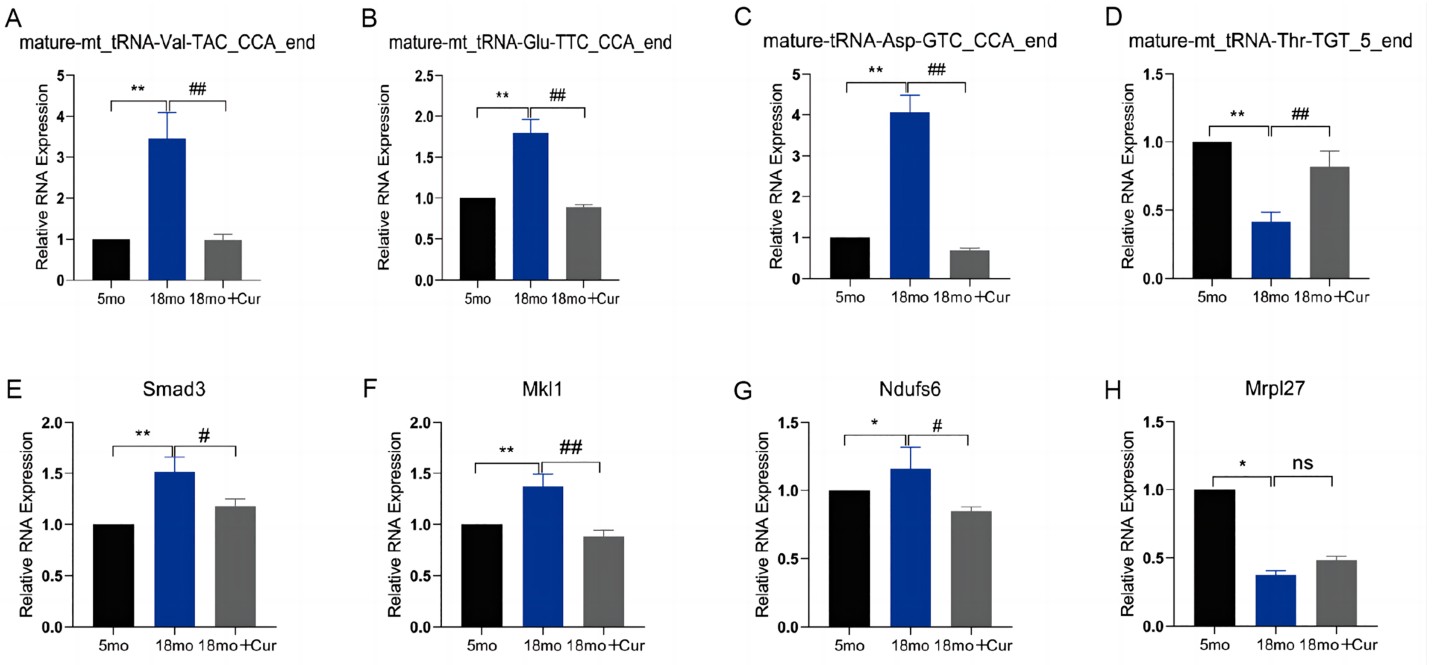

**Figure 5 qRT-PCR verification of differential tsRNA and target genes *in vitro*.** (A) The expression of mature-mt_tRNA-Val-TAC_CCA_end. (B) The expression of tsRNAs mature-mt_tRNA-Thr-TGT_5_end. (C) The expression of mature-mt_tRNA-Glu-TTC_CCA_end. (D) The expression of tsRNAs mature-tRNA-Asp-GTC_CCA_end. (E) The expression of Smad3. (F) The expression of Mkl1. (G) The expression of Ndufs6. (H) The expression of Mrpl27. TsRNA expression normalized to U6, mRNA expression normalized to GAPDH, $n = 6$, *$P < 0.05$, **$P < 0.01$ vs. 5 month group; #$P < 0.05$, ##$P < 0.01$ vs. 18 month group.

## Mature-mt_tRNA-Val-TAC_CCA_end regulated the celluar senecence and the levels of ROS and inflammatory factors

TsRNA mimics and inhibitors were transfected into MAF to verify the target mRNA expression levels of tsRNAs. Subsequently, mature-mt_tRNA-Val-TAC_CCA_end was selected for further verification as it contained the most target genes. We found that mature-mt_tRNA-Val-TAC_CCA_end mimic treated group had more senescent cells, increased ROS levels and IL-6/TNFα expression compared with the control group (Figs. 6A–6C). The findings indicate that overexpression of mature- mt_tRNA-Val-TAC_CCA_end can induce cellular senescence, oxidative stress, and elevated levels of inflammation. Compared with the TBHP group, the inhibitor group showed fewer senescent cells, decreased ROS levels and IL-6/TNFα expression, suggesting that inhibiting mature-mt_tRNA-Val-TAC_CCA_end can reduce the ratio of senescent cells, oxidative stress, and inflammation levels (Figs. 7A–7C). Moreover, increased expression of the core target genes Smad3, Ndufs6, Mkl1, and Mrpl27 was observed in the mimic group, while decreased gene expression in the inhibitor group compared with the control group excepting for Mrpl27. These results suggest that mature-mt_tRNA-Val-TAC_CCA_end may regulate Smad3, Ndufs6, Mkl1, and Mrpl27, which are involved in the process of aging, fibrosis, oxidation and inflammation.

## DISCUSSION

AF is the most common sustained cardiac arrhythmia, and age has been shown to be an independent risk factor for AF. Growing evidence shows that intrinsic stress, such as oxidative stress and chronic inflammation, is significantly enhanced in aging and AF, and the aging heart shows a higher susceptibility to extrinsic stress stimuli (*Verma & Wong, 2019*). As a "novel star" drug for the treatment of age-related diseases, curcumin can delay cellular aging and reduce aging-related phenotypes. However, it is not clear whether curcumin improves aging-related AF, and its drug target also remains exploring. According to published literature on rodents (including rats and mice), the dose of curcumin (100 mg/kg/d) used in this study is effective and safe (*Moulin et al., 2020*; *Topcu-Tarladacalisir et al., 2013*). And in our previous research, we also found that curcumin (100 mg/kg/d, oral) improved atrial fibrosis, and decreased the susceptibility of AF in aged mice when compared to 50 mg/kg/day of curcumin and control group (*Gao et al., 2023*). So in this study, we used curcumin (100 mg/kg/day, oral) to predict its mechanism in the treatment of aging-related AF by regulating tsRNAs, and verified those results with *in vitro* and in-vivo experiments. The results demonstrated that curcumin reduced susceptibility to AF in aged mice. A large number of sncRNAs were found in the atrium tissues of three groups, among which tsRNA and rsRNA accounted for the highest proportion. The tsRNAs were then selected for general characteristics and bioinformatics analysis. According to functional enrichment analysis, the target genes of differential tsRNAs were mainly involved in processes such as DNA damage, protein modification, and nutrient metabolism. Moreover, overexpression or inhibitorn of mature-mt_tRNA-Val-TAC_CCA could positively and negatively regulate the levels of oxidation-inflammation-aging. This

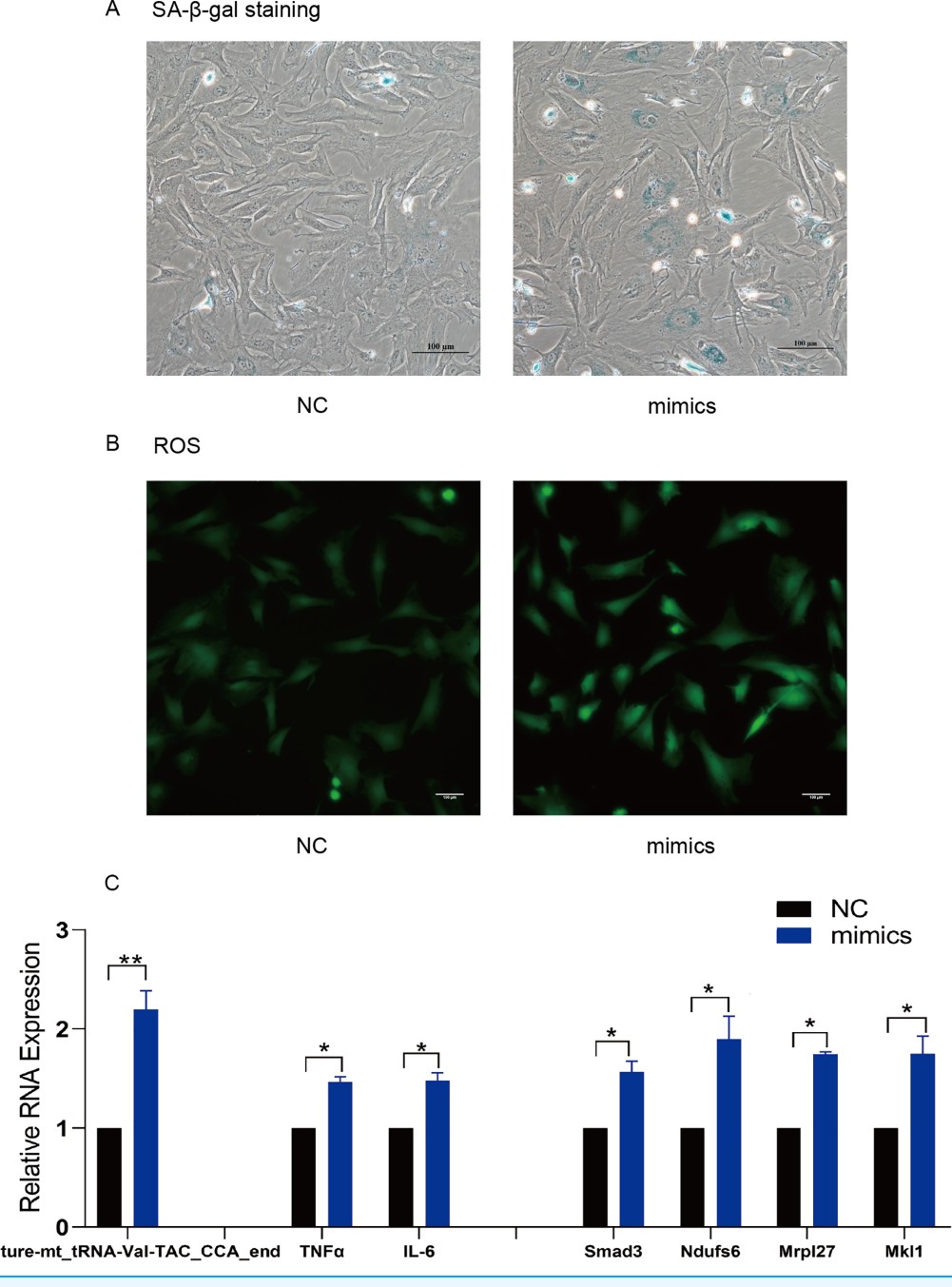

**Figure 6 Overexpression of mature-mt_tRNA-Val-TAC_CCA_end upregulated the level of oxidation-inflammation-aging.** (A) The ratio of senescent cells after treated with mature-mt_tRNA-Val-TAC_CCA_end mimics. Scale bar, 100 μm. (B) ROS detection. Scale bar, 100 μm. (C) The expression of inflammation indicators and target genes. $n$ = 3–4, *$P < 0.05$ vs. NC, **$P < 0.01$ vs. NC.

suggests that curcumin reduces oxidative stress, the inflammatory response and aging effects by regulating tsRNAs, thereby reducing the susceptibility to AF in aged mice.

TsRNA is a new type of small non-coding RNA of epigenetic regulatory factors, and is involved in physiological and pathological processes such as gene expression, cell

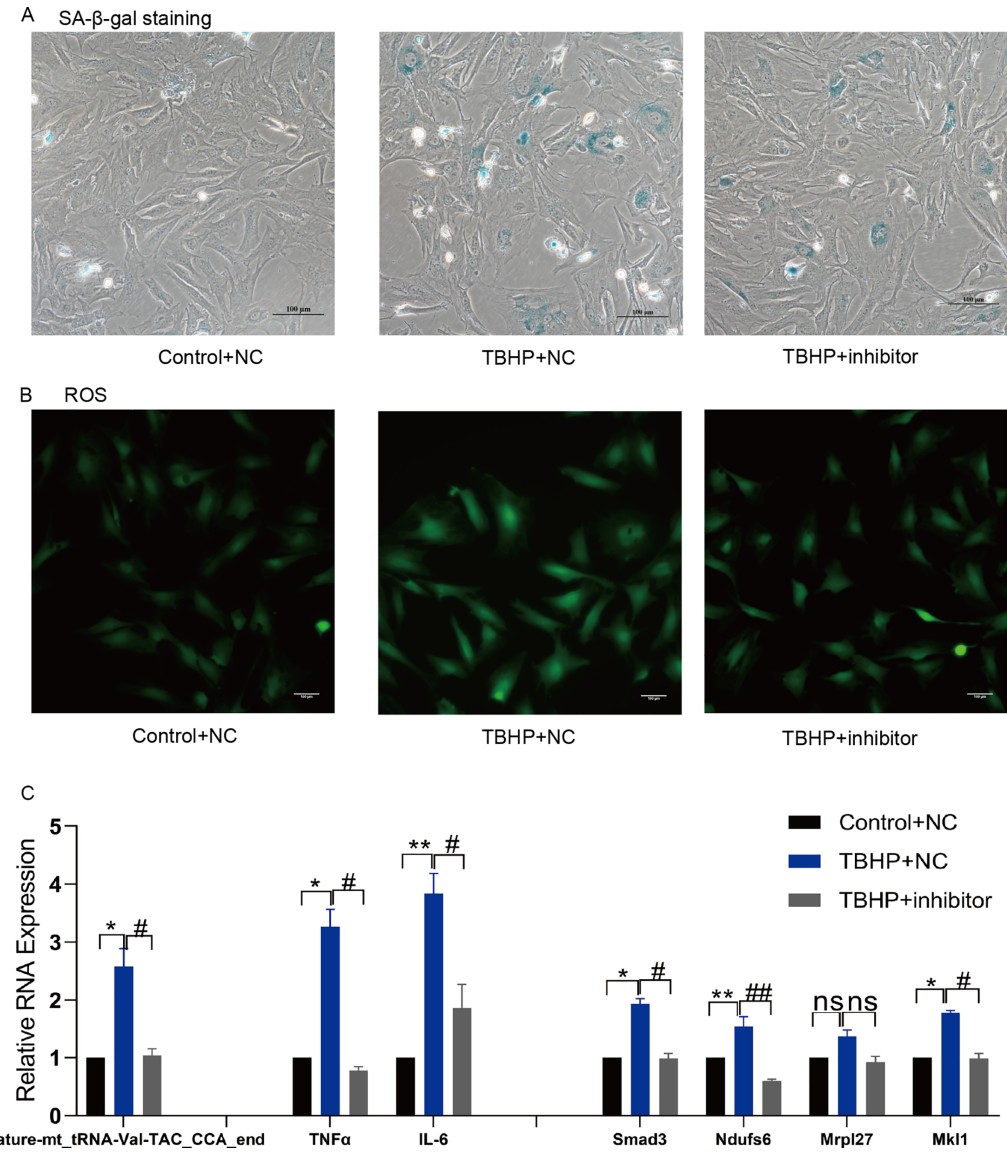

**Figure 7 Inhibition of mature-mt_tRNA-Val-TAC_CCA_end improved oxidation-inflammation-aging.** (A) The ratio of senescent cells after treated with mature-mt_tRNA-Val-TAC_CCA_end inhibitor. Scale bar, 100 μm. (B) ROS detection. Scale bar, 100 μm. (C) The expression of inflammation indicators and target genes. $n = 3–4$, $*P < 0.05$, $**P < 0.01$ vs. Control + NC; $^{\#}P < 0.05$, $^{\#\#}P < 0.01$ vs. TBHP+NC.

apoptosis, and epigenetics. A very important function of tsRNA is to regulate gene expression by binding to mRNA in a sequence-specific manner similar to miRNA (*Zhang et al., 2020*). Although the total miRNA level gradually decreases during aging, the tsRNA level of Cryptomeria elegans increases during aging, which may be a response to aging induced stress (*Kim & Lee, 2019*). In this study, SPORTS1.1 was used to annotate the expression profiles of tsRNAs in atrial tissue of young (5 month) and aged mice (18 month) treated with or without curcumin. According to the tRNA regions from which the tsRNAs are derived, tsRNAs can be divided into four groups: 3′-end, 5′-end, CCA-end,

and inter-tsRNA. tsRNAs are mainly composed of glycine (Gly), serine (Ser), valine (Val), and glutamine (Glu) tRNA derivatization. And significant differences in origin loci were observed among the three groups. Moreover, the total tsRNA expression was upregulated in the 18 month group compared to the 5 month control mice, which was reversed by the treatment of curcumin. The expression trend of tsRNA-Glu and tsRNA-Val was consistent with that of total tsRNA, while tsRNA-Gly showed an opposite trend, indicating that different subtypes of tsRNA responded differently to aging-related stress. In summary, aging leads to changes in the expression profile of atrial appendage tsRNA, and curcumin might inhibit the progression of aging-related AF by targeting tsRNA.

Through the differential expression analysis of the tsRNA expression profile, 99 common differentially expressed tsRNA sequences (12 kinds of tsRNAs species) between the pairwise comparisons in these three groups. More than 50% of the differential tsRNA sequences originated from mitochondrial tRNAs. Cardiac aging is closely associated with mitochondrial dysfunction (*Li et al., 2020a*). Mitochondria-to-nucleus (silk-mitochondrion-nucleus) bidirectional communication causes various epigenetic changes through altered levels of metabolites or stress signals and participates in the aging process (*Sharma, Pasala & Prakash, 2019*). Emerging research suggests that multiple mitochondrial RNAs (mtRNAs: mt-rRNA, mt-tRNA, mt-mRNA, and multiple non-coding RNAs) present in the mitochondrial matrix are also involved in epigenetic regulation (*Dong, Pu & Cui, 2020*). In addition, ROS induces tRNA cleavage into tsRNA by regulating its biogenesis pathway, and mitochondria are the main source of ROS production (*Wagner et al., 2022*). Therefore, curcumin might exert anti-aging effects through acting on mitochondria, and tsRNA might be its targeted epigenetic factor.

Functional enrichment analysis was performed to study the functions of tsRNA targets, and the target genes were mainly involved in processes such as DNA damage, protein modification, nutrient metabolism, and other processes, including SUMOylation of transcription cofactors, and 2-Oxocarboxylic acid metabolism, *etc*. Emerging evidence suggests that DNA damage response (DDR) signaling is a key mechanism linking DNA damage accumulation, cellular senescence, and organism aging (*Olivieri et al., 2015*). DNA damage is an important feature of aging, and chronic inflammation and elevated levels of ROS are important environmental factors of DNA damage (*Das et al., 2023*). DDR activation in senescent cells promotes senescence-associated secretory phenotype (SASP), which in turn causes DDR and SASP activation in neighboring cells (*Olivieri et al., 2015*). In addition to nuclear DNA, mitochondrial DNA damage can also accelerate aging. A study found that homozygous mutations in mitochondrial polymerase gamma (Polgm/m) in mice and accumulation of mtDNA mutations and deletions caused cardiac hypertrophy and accelerated cardiac aging (*Trifunovic et al., 2004*). SIRT1/3 activation has been reported to ameliorate age-induced cardiac dysfunction by repairing mtDNA damage, suggesting the therapeutic potential of SIRT1/3 targeting mtDNA in cardiac aging (*Marin-Garcia, 2016*; *Quan et al., 2020*). SUMO is a small ubiquitin-like protein but acts opposite to ubiquitin, and both of which can occur under oxidative stress. Scientists found that the Lys734 site of SIRT1 was SUMOylated in senescent cells, and in response to oxidative

stress and DNA damage, SIRT1 combined with SENP1 and SIRT1 was deSUMOlated and its activity was decreased (*Flick & Luscher, 2012*; *Yang et al., 2007*). Therefore, oxidative stress can cause a series of DNA damage and epigenetic modification at the protein level to further promote aging-related phenotypes.

Additionally, in the present study, we found that mature-mt_tRNA-Val-TAC_CCA_end, mature-mt_tRNA-Glu-TTC_CCA_end, and mature-tRNA-Asp-GTC_CCA_end were up-regulated in the 18 month group compared with the 5 month group, while mature-mt_tRNA-Thr-TGT_5_end was down-regulated, which could be reversed by curcumin. Moreover, activation of mature-mt_tRNA-Val-TAC_CCA_end was observed to increase the level of oxidative stress/inflammation and promote cell aging. At the same time, the core target genes Smad3, Mkl1, Ndufs6, and Mrpl27 of mature-mt_tRNA-Val-TAC_CCA_end all participated in the process of oxidation, inflammation, and aging. Recent studies have found that Smad3 and Mkl1 are also regulated by miRNAs (*Hampl et al., 2013*; *Yang et al., 2018*). Moreover, our previous study also found that Smad3 was involved in the process of aging-related atrial fibrosis (*Gao et al., 2023*). Mrp127 is involved in the expression of proteins encoded by the mitochondrial genome, and the expression of Mrp127 has been reported to be up-regulated in metabolic syndrome (MS) rat hearts (*Zheng et al., 2017*). Furthermore, Ndufs6 is closely related to mitochondrial function, and abnormal expression can lead to excessive ROS accumulation and upregulation of p53/p21 to promote the aging process (*Zhang et al., 2020*). Unlike Smad3, Mkl1 and Ndufs6, there was no significant change of Mrpl27 expression after inhibition of mature-mt_tRNA-Val-TAC_CCA_end. The mature-mt_tRNA-Val-TAC_CCA_end might affect the expression level of mRNA by regulating RNA binding protein (RBP) with its target genes Smad 3, Mkl1 and Ndufs6. However, as Mrpl27 mRNA did not change significantly, it was likely that mature-mt_tRNA-Val-TAC_CCA_end induced a conformational change of mRNA that enhanced the translation process without affecting the expression of mRNA. However, The above results confirmed that mature-mt_tRNA-Val-TAC_CCA_end can regulate target genes to promote the aging process through the oxidation-inflammation-aging network, which is consistent with our bioinformatics analysis. These results indicated that tsRNA may be one of the anti-aging molecules of curcumin, with mature-mt_tRNA-Val-TAC_CCA_end being the drug target.

Nevertheless, the limitations of the present study should be acknowledged. First, future studies with larger sample sizes and including clinical samples are required to confirm the results. Second, the specific interactions and binding sites between tsRNAs and mRNAs require further investigation. Finally, tsRNAs are closely related to mitochondrial function, and their mitochondrial biochemical reactions can be further elucidated.

## CONCLUSION

In summary, the role of tsRNAs in the anti-aging properties of curcumin has rarely been reported. Our study reveals changes in the expression profile of tsRNAs in curcumin-treated aged atria and suggests the potential therapeutic mechanism of tsRNAs in aged AF. This study may encourage further research on the regulatory role of tsRNAs in aging-related AF.

### Funding
This work was supported by the High-level Hospital Construction Plan (Nos. DFJH201808 and DFJH201925), the Guangdong special funds for science and technology innovation strategy, China (Stability support for scientific research institutions affiliated to Guangdong Province-GDCI 2021) and the Science and Technology Program of Guangdong Province (No. 2019B020230004). The funders had no role in study design, data collection and analysis, decision to publish, or preparation of the manuscript.

### Grant Disclosures
The following grant information was disclosed by the authors:
High-level Hospital Construction Plan: DFJH201808 and DFJH201925.
Science and Technology Innovation Strategy, China: GDCI 2021.
Science and Technology Program of Guangdong Province: 2019B020230004.

### Competing Interests
The authors declare that they have no competing interests.

### Author Contributions
- Xueshan Luo conceived and designed the experiments, prepared figures and/or tables, and approved the final draft.
- Panyue Liu conceived and designed the experiments, performed the experiments, prepared figures and/or tables, and approved the final draft.
- Xingdong Ye performed the experiments, prepared figures and/or tables, and approved the final draft.
- Jintao He performed the experiments, prepared figures and/or tables, and approved the final draft.
- Yingyu Lai performed the experiments, authored or reviewed drafts of the article, and approved the final draft.
- Yidong Lv analyzed the data, prepared figures and/or tables, and approved the final draft.
- Xiongbin Wu conceived and designed the experiments, analyzed the data, authored or reviewed drafts of the article, and approved the final draft.
- Yang Liu analyzed the data, authored or reviewed drafts of the article, and approved the final draft.
- Qianhuan Zhang analyzed the data, authored or reviewed drafts of the article, and approved the final draft.
- Hui Yang performed the experiments, prepared figures and/or tables, authored or reviewed drafts of the article, and approved the final draft.
- Wei Wei performed the experiments, prepared figures and/or tables, authored or reviewed drafts of the article, and approved the final draft.
- Chunyu Deng analyzed the data, prepared figures and/or tables, authored or reviewed drafts of the article, and approved the final draft.

- Sujuan Kuang performed the experiments, authored or reviewed drafts of the article, and approved the final draft.
- Shulin Wu analyzed the data, authored or reviewed drafts of the article, and approved the final draft.
- Yumei Xue analyzed the data, authored or reviewed drafts of the article, and approved the final draft.
- Fang Rao conceived and designed the experiments, analyzed the data, authored or reviewed drafts of the article, and approved the final draft.

## Animal Ethics

The following information was supplied relating to ethical approvals (*i.e.*, approving body and any reference numbers):

All procedures and protocols in this study have been approved by the animal experiment ethics committee of Guangdong Provincial People's Hospital, Guangdong Academy of Medical Sciences (ethics number GDREC2016128A).

## Data Availability

The data is available in the Supplemental File and at NCBI GEO: GSE217761.

## Supplemental Information

Supplemental information for this article can be found online at http://dx.doi.org/10.7717/peerj.17495#supplemental-information.

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
