# Peer review of "Curcumin improves atrial fibrillation susceptibility by regulating tsRNA expression in aging mouse atrium"

_PeerJ, doi:10.7717/peerj.17495_

## Round 0.1 · original submission · Major Revisions

I suggest to carefully consider all the Reviewers' comments.

Reviewer 1 ·

Basic reporting

Luo et al. delve into the role of curcumin in addressing aging-related atrial fibrillation (AF) by modulating tsRNAs, corroborating their findings with in vitro and in vivo experiments. Their focus on tsRNAs, a novel class of short non-coding RNAs, is pioneering and could stimulate significant research interest, furthering our understanding of their potential functionalities.

Experimental design

While high-throughput RNA sequencing highlighted differential tsRNAs expression across the three experimental groups, most in vitro studies were limited to the exploration of a single tsRNA, mature-mt_tRNA-Val-TAC_CCA_end.

Validity of the findings

Drawing conclusions about this specific tsRNA as a predictive marker without evaluating other tsRNAs' expression in mouse atrial tissue and atrial fibroblasts seems rather restrictive.

Additional comments

The following specific comments in the attachment require further elaboration.

Annotated reviews are not available for download in order to protect the identity of reviewers who chose to remain anonymous.

Reviewer 2 ·

Basic reporting

Unambiguous, professional English is used throughout.
Literature references and sufficient field background/context provided.
Acceptable format.
Figures and tables with a sufficient resolution and appropriately described and labeled.

Experimental design

Research question well defined, methods well described.

Validity of the findings

Excellent

Additional comments

No additional comments

Reviewer 3 ·

Basic reporting

Authors have succinctly laid out the context of this study and the hypotheses tested. While the authors have presented the results clearly, there are some areas that need to be improved upon both in terms of language and figures presented. These are noted below:
• Authors need to revise the figures; the aspect ratio is distorted in several instances and the text sizes (e.g. graph axes labels and legends) across figures vary significantly.
• Global comment: The figure legends are very sparse and do not effectively communicate the content in the figure. Authors should be more comprehensive in mentioning what is described in the figure, the sample, n sizes used, stats, etc.
• Line 26: Delete “the”; change to “reducing the aging-related oxidative stress and inflammation”
• Line 33-35/ Line 222, 224: The authors write that “Compared to young mice (5month) group, aged mice yielded 473 significantly dysregulated tsRNA sequences, while 947 tsRNA sequences were significantly dysregulated after treated with curcumin.” Can the authors change the word “dysregulated” to “altered” wherever applicable? Dysregulation eludes to impairment of a physiological regulatory mechanism, however, in this context, it cannot be concluded that all these changes are a result of some impaired mechanism.
• Line 50-51: “Due to the aging population, the burden of AF is rapidly increasing worldwide and is expected to at least triple over the next 50 years.” Please add the source for this statistic.
• Line 65: “Epigenetic alterations are considered a hallmark of aging”
• Line 207: “tsRNA and rsRNA accounted 207 for the largest proportion of changes.”
• Suggesting authors use a different color pattern for Figures 2 and 3. Since in Figure 3, subgroups of tsRNAs are being analyzed, authors should use different colors that do not overlap with sncRNAs analyzed in Figure 2.
• Line 238: “Compared with the 5mo group, qRT-PCR results demonstrated….”

Experimental design

The research question is well defined and authors have done a fair job to design experiments to validate their hypotheses. However, there are some outstanding questions that the authors absolutely need to address:
• Authors should expand on the findings, especially in the results for “Exemplary annotation and profiling of PANDORA-seq datasets generated by SPORTS1.1”. This is a rich dataset and the authors can include more information on the numbers of modified sncRNAs, how they change across the different treatment groups. Authors should provide a quantitative estimate of the proportion of tsRNA and rsRNA that have changed in the sample.
-Authors should use the same axes for Figure 2B as for 2A and 2C. Though the authors claim
that there is no significant difference in total expression observed among the three treatment
groups, the authors should back it up with quantitative data.
-How is the proportion of tsRNAs with respect to all the other sncRNAs different in the different
treatment groups?
-Authors note that the sncRNAs were concentrated in lengths ranging from 15-45 nt. Is there
any implication of this finding?
• Authors should keep the y-axes range consistent in Figures 3A, B and C for easier comparison of overall numbers.
-A similar comment for Figure G, H and I. A suggestion to authors to overlay these figures
across different Tx groups for easier comparison?
• Line 222-229: Authors should present analysis of the sncRNAs / tsRNAs that were altered (both up and downregulated) at 18 months compared to 5 months. And how many of those reversed their pattern in 18mo+curcumin sample? Given this is the basis of the study, authors should present these findings very clearly. The bottom set of genes in Figure 4B i.e. those that are upregulated in 18 mo but stay lower in the 18 mo + curcumin sample is particularly interesting and the authors should comment on those specifically. (this analysis should in turn also show all the genes presented in Figure 5A-D
-It is also unclear what the overlap in Fig 4D is conveying

Validity of the findings

While the data presented is largely robust, there are some missing links in the story that need to be justified by the authors.
• It is unclear how and why the authors have “randomly” selected 4 tsRNAs to explore their function. Authors should mention this more clearly.
• Figure 4E: Authors should clarify in the main results section what the Cytoscape result yielded and how that tie with the tsRNA target genes on metascape. The results presented are disconnected and poorly described.
• Line 243-244: Authors should comment on why Mrpl27 expression did not follow the same pattern as Smad3, Mkl1, Ndufs6 and underlying possible causes
• Line 249-250: Authors note that “mature-mt_tRNA-Val-TAC_CCA_end was selected for further verification as it contained the most target genes”
- Authors should comment on the number of target genes for each of the tsRNAs tested. Is the n
number of target genes significantly different to truly justify why mature-mt_tRNA-Val-
TAC_CCA_end was selected for further analysis?

---

## Round 0.2 · Minor Revisions

Please clarify how you chose the dose of curcumin. Please provide a reference for this (at least one study that showed that dose, administered orally provided effects). Did you do preliminary experiments to assess the dose?

Reviewer 1 ·

Basic reporting

The authors have addressed all my comments.

Experimental design

The manuscript is ready for publication.

Validity of the findings

See above.

---

## Round 0.3 · Minor Revisions

The Authors answered my questions but the Section Editor noted:

> Could the authors please elaborate on group sizes and biological/technical replicates in all figures, especially in the PANDORA data set? They comment on statistical power, so I don't doubt the veracity of this, but I'd like to see the numbers specified.

---

## Round 0.4 · accepted · Accept

The requested corrections were made.